# Online Learning with Switching Costs and Other Adaptive Adversaries

**Nicolò Cesa-Bianchi**
Università degli Studi di Milano
Italy

**Ofer Dekel**
Microsoft Research
USA

**Ohad Shamir**
Microsoft Research
and the Weizmann Institute

## Abstract

We study the power of different types of adaptive (nonoblivious) adversaries in the setting of prediction with expert advice, under both full-information and bandit feedback. We measure the player's performance using a new notion of regret, also known as policy regret, which better captures the adversary's adaptiveness to the player's behavior. In a setting where losses are allowed to drift, we characterize —in a nearly complete manner— the power of adaptive adversaries with bounded memories and switching costs. In particular, we show that with switching costs, the attainable rate with bandit feedback is $\widetilde{\Theta}(T^{2/3})$. Interestingly, this rate is significantly worse than the $\Theta(\sqrt{T})$ rate attainable with switching costs in the full-information case. Via a novel reduction from experts to bandits, we also show that a bounded memory adversary can force $\widetilde{\Theta}(T^{2/3})$ regret even in the full information case, proving that switching costs are easier to control than bounded memory adversaries. Our lower bounds rely on a new stochastic adversary strategy that generates loss processes with strong dependencies.

## 1 Introduction

An important instance of the framework of prediction with expert advice —see, e.g., [8]— is defined as the following repeated game, between a randomized player with a finite and fixed set of available actions and an adversary. At the beginning of each round of the game, the adversary assigns a loss to each action. Next, the player defines a probability distribution over the actions, draws an action from this distribution, and suffers the loss associated with that action. The player's goal is to accumulate loss at the smallest possible rate, as the game progresses. Two versions of this game are typically considered: in the *full-information feedback* version, at the end of each round, the player observes the adversary's assignment of loss values to each action. In the *bandit feedback* version, the player only observes the loss associated with his chosen action, but not the loss values of other actions.

We assume that the adversary is *adaptive* (also called *nonoblivious* by [8] or *reactive* by [16]), which means that the adversary chooses the loss values on round $t$ based on the player's actions on rounds $1 \ldots t-1$. We also assume that the adversary is deterministic and has unlimited computational power. These assumptions imply that the adversary can specify his entire strategy before the game begins. In other words, the adversary can perform all of the calculations needed to specify, in advance, how he plans to react on each round to any sequence of actions chosen by the player.

More formally, let $\mathcal{A}$ denote the finite set of actions and let $X_t$ denote the player's random action on round $t$. We adopt the notation $X_{1:t}$ as shorthand for the sequence $X_1 \ldots X_t$. We assume that the adversary defines, in advance, a sequence of *history-dependent loss functions* $f_1, f_2, \ldots$. The input to each loss function $f_t$ is the entire history of the player's actions so far, therefore the player's loss on round $t$ is $f_t(X_{1:t})$. Note that the player doesn't observe the functions $f_t$, only the losses that result from his past actions. Specifically, in the bandit feedback model, the player observes $f_t(X_{1:t})$ on round $t$, whereas in the full-information model, the player observes $f_t(X_{1:t-1}, x)$ for all $x \in \mathcal{A}$.

On any round $T$, we evaluate the player's performance so far using the notion of *regret*, which compares his cumulative loss on the first $T$ rounds to the cumulative loss of the best fixed action in hindsight. Formally, the player's regret on round $T$ is defined as

$$R_T = \sum_{t=1}^{T} f_t(X_{1:t}) - \min_{x \in \mathcal{A}} \sum_{t=1}^{T} f_t(x \ldots x) \ . \tag{1}$$

$R_T$ is a random variable, as it depends on the randomized action sequence $X_{1:t}$. Therefore, we also consider the *expected regret* $\mathbb{E}[R_T]$. This definition is the same as the one used in [18] and [3] (in the latter, it is called *policy regret*), but differs from the more common definition of expected regret

$$\mathbb{E}\left[\sum_{t=1}^{T} f_t(X_{1:t}) - \min_{x \in \mathcal{A}} \sum_{t=1}^{T} f_t(X_{1:t-1}, x)\right] \ . \tag{2}$$

The definition in Eq. (2) is more common in the literature (e.g., [4, 17, 10, 16]), but is clearly inadequate for measuring a player's performance against an adaptive adversary. Indeed, if the adversary is adaptive, the quantity $f_t(X_{1:t-1}, x)$ is hardly interpretable —see [3] for a more detailed discussion.

In general, we seek algorithms for which $\mathbb{E}[R_T]$ can be bounded by a sublinear function of $T$, implying that the per-round expected regret, $\mathbb{E}[R_T]/T$, tends to zero. Unfortunately, [3] shows that arbitrary adaptive adversaries can easily force the regret to grow linearly. Thus, we need to focus on (reasonably) weaker adversaries, which have constraints on the loss functions they can generate.

The weakest adversary we discuss is the *oblivious adversary*, which determines the loss on round $t$ based only on the current action $X_t$. In other words, this adversary is oblivious to the player's past actions. Formally, the oblivious adversary is constrained to choose a sequence of loss functions that satisfies $\forall t$, $\forall x_{1:t} \in \mathcal{A}^t$, and $\forall x'_{1:t-1} \in \mathcal{A}^{t-1}$,

$$f_t(x_{1:t}) = f_t(x'_{1:t-1}, x_t) \ . \tag{3}$$

The majority of previous work in online learning focuses on oblivious adversaries. When dealing with oblivious adversaries, we denote the loss function by $\ell_t$ and omit the first $t-1$ arguments. With this notation, the loss at time $t$ is simply written as $\ell_t(X_t)$.

For example, imagine an investor that invests in a single stock at a time. On each trading day he invests in one stock and suffers losses accordingly. In this example, the investor is the player and the stock market is the adversary. If the investment amount is small, the investor's actions will have no measurable effect on the market, so the market is oblivious to the investor's actions. Also note that this example relates to the full-information feedback version of the game, as the investor can see the performance of each stock at the end of each trading day.

A stronger adversary is the *oblivious adversary with switching costs*. This adversary is similar to the oblivious adversary defined above, but charges the player an additional switching cost of 1 whenever $X_t \neq X_{t-1}$. More formally, this adversary defines his sequence of loss functions in two steps: first he chooses an oblivious sequence of loss functions, $\ell_1, \ell_2 \ldots$, which satisfies the constraint in Eq. (3). Then, he sets $f_1(x) = \ell_1(x)$, and

$$\forall \, t \geq 2, \quad f_t(x_{1:t}) = \ell_t(x_t) + \mathbb{I}_{\{x_t \neq x_{t-1}\}} \ . \tag{4}$$

This is a very natural setting. For example, let us consider again the single-stock investor, but now assume that each trade has a fixed commission cost. If the investor keeps his position in a stock for multiple trading days, he is exempt from any additional fees, but when he sells one stock and buys another, he incurs a fixed commission. More generally, this setting (or simple generalizations of it) allows us to capture any situation where choosing a different action involves a costly change of state. In the paper, we will also discuss a special case of this adversary, where the loss function $\ell_t(x)$ for each action is sampled i.i.d. from a fixed distribution.

The switching costs adversary defines $f_t$ to be a function of $X_t$ and $X_{t-1}$, and is therefore a special case of a more general adversary called an *adaptive adversary with a memory of* 1. This adversary is constrained to choose loss functions that satisfy $\forall t$, $\forall x_{1:t} \in \mathcal{A}^t$, and $\forall x'_{1:t-2} \in \mathcal{A}^{t-2}$,

$$f_t(x_{1:t}) = f_t(x'_{1:t-2}, x_{t-1}, x_t) \ . \tag{5}$$

This adversary is more general than the switching costs adversary because his loss functions can depend on the previous action in an arbitrary way. We can further strengthen this adversary and

define the *bounded memory adaptive adversary*, which has a bounded memory of an arbitrary size. In other words, this adversary is allowed to set his loss function based on the player's $m$ most recent past actions, where $m$ is a predefined parameter. Formally, the bounded memory adversary must choose loss functions that satisfy, $\forall t, \forall x_{1:t} \in \mathcal{A}^t$, and $\forall x'_{1:t-m-1} \in \mathcal{A}^{t-m-1}$,

$$ f_t(x_{1:t}) \;=\; f_t(x'_{1:t-m-1}, x_{t-m:t}) \;\;. $$

In the information theory literature, this setting is called *individual sequence prediction against loss functions with memory* [18].

In addition to the adversary types described above, the bounded memory adaptive adversary has additional interesting special cases. One of them is the *delayed feedback oblivious adversary* of [19], which defines an oblivious loss sequence, but reveals each loss value with a delay of $m$ rounds. Since the loss at time $t$ depends on the player's action at time $t - m$, this adversary is a special case of a bounded memory adversary with a memory of size $m$. The delayed feedback adversary is not a focus of our work, and we present it merely as an interesting special case.

So far, we have defined a succession of adversaries of different strengths. This paper's goal is to understand the upper and lower bounds on the player's regret when he faces these adversaries. Specifically, we focus on how the expected regret depends on the number of rounds, $T$, with either full-information or bandit feedback.

## 1.1 The Current State of the Art

Different aspects of this problem have been previously studied and the known results are surveyed below and summarized in Table 1. Most of these previous results rely on the additional assumption that the range of the loss functions is bounded in a fixed interval, say $[0, C]$. We explicitly make note of this because our new results require us to slightly generalize this assumption.

As mentioned above, the oblivious adversary has been studied extensively and is the best understood of all the adversaries discussed in this paper. With full-information feedback, both the *Hedge* algorithm [15, 11] and the *follow the perturbed leader (FPL)* algorithm [14] guarantee a regret of $\mathcal{O}(\sqrt{T})$, with a matching lower bound of $\Omega(\sqrt{T})$ —see, e.g., [8]. Analyses of Hedge in settings where the loss range may vary over time have also been considered —see, e.g., [9]. The oblivious setting with bandit feedback, where the player only observes the incurred loss $f_t(X_{1:t})$, is called the *nonstochastic (or adversarial) multi-armed bandit* problem. In this setting, the Exp3 algorithm of [4] guarantees the same regret $\mathcal{O}(\sqrt{T})$ as the full-information setting, and clearly the full-information lower bound $\Omega(\sqrt{T})$ still applies.

The *follow the lazy leader (FLL)* algorithm of [14] is designed for the switching costs setting with full-information feedback. The analysis of FLL guarantees that the oblivious component of the player's expected regret (without counting the switching costs), as well as the expected number of switches, is upper bounded by $\mathcal{O}(\sqrt{T})$, implying an expected regret of $\mathcal{O}(\sqrt{T})$.

The work in [3] focuses on the bounded memory adversary with bandit feedback and guarantees an expected regret of $\mathcal{O}(T^{2/3})$. This bound naturally extends to the full-information setting. We note that [18, 12] study this problem in a different feedback model, which we call *counterfactual feedback*, where the player receives a full description of the history-dependent function $f_t$ at the end of round $t$. In this setting, the algorithm presented in [12] guarantees an expected regret of $\mathcal{O}(\sqrt{T})$.

Learning with bandit feedback and switching costs has mostly been considered in the economics literature, using a different setting than ours and with prior knowledge assumptions (see [13] for an overview). The setting of stochastic oblivious adversaries (i.e., oblivious loss functions sampled i.i.d. from a fixed distribution) was first studied by [2], where they show that $\mathcal{O}(\log T)$ switches are sufficient to asymptotically guarantee logarithmic regret. The paper [20] achieves logarithmic regret nonasymptotically with $\mathcal{O}(\log T)$ switches.

Several other papers discuss online learning against "adaptive" adversaries [4, 10, 16, 17], but these results are not relevant to our work and can be easily misunderstood. For example, several bandit algorithms have extensions to the "adaptive" adversary case, with a regret upper bound of $\mathcal{O}(\sqrt{T})$ [1]. This bound doesn't contradict the $\Omega(T)$ lower bound for general adaptive adversaries mentioned

| | oblivious | switching cost | memory of size 1 | bounded memory | adaptive |
|---|---|---|---|---|---|
| | | | Full-Information Feedback | | |
| $\widetilde{\mathcal{O}}$ | $\sqrt{T}$ | $\sqrt{T}$ | $T^{2/3}$ | $T^{2/3}$ | $T$ |
| $\Omega$ | $\sqrt{T}$ | $\sqrt{T}$ | $\sqrt{T}$ | $\sqrt{T} \rightarrow \boldsymbol{T^{2/3}}$ | $T$ |
| | | | Bandit Feedback | | |
| $\widetilde{\mathcal{O}}$ | $\sqrt{T}$ | $T^{2/3}$ | $T^{2/3}$ | $T^{2/3}$ | $T$ |
| $\Omega$ | $\sqrt{T}$ | $\sqrt{T} \rightarrow \boldsymbol{T^{2/3}}$ | $\sqrt{T} \rightarrow \boldsymbol{T^{2/3}}$ | $\sqrt{T} \rightarrow \boldsymbol{T^{2/3}}$ | $T$ |

Table 1: State-of-the-art upper and lower bounds on regret (as a function of $T$) against different adversary types. Our contribution to this table is presented in bold face.

earlier, since these papers use the regret defined in Eq. (2) rather than the regret used in our work, defined in Eq. (1).

Another related body of work lies in the field of competitive analysis —see [5], which also deals with loss functions that depend on the player's past actions, and the adversary's memory may even be unbounded. However, obtaining sublinear regret is generally impossible in this case. Therefore, competitive analysis studies much weaker performance metrics such as the competitive ratio, making it orthogonal to our work.

## 1.2   Our Contribution

In this paper, we make the following contributions (see Table 1):

- Our main technical contribution is a new lower bound on regret that matches the existing upper bounds in several of the settings discussed above. Specifically, our lower bound applies to the switching costs adversary with bandit feedback and to all strictly stronger adversaries.
- Building on this lower bound, we prove another regret lower bound in the bounded memory setting with *full-information* feedback, again matching the known upper bound.
- We confirm that existing upper bounds on regret hold in our setting and match the lower bounds up to logarithmic factors.
- Despite the lower bound, we show that for switching costs and bandit feedback, if we also assume *stochastic i.i.d. losses*, then one can get a distribution-free regret bound of $\mathcal{O}(\sqrt{T \log \log \log T})$ for finite action sets, with only $\mathcal{O}(\log \log T)$ switches. This result uses ideas from [7], and is deferred to the supplementary material.

Our new lower bound is a significant step towards a complete understanding of adaptive adversaries; observe that the upper and lower bounds in Table 1 essentially match in all but one of the settings.

Our results have two important consequences. First, observe that the optimal regret against the switching costs adversary is $\Theta(\sqrt{T})$ with full-information feedback, versus $\Theta(T^{2/3})$ with bandit feedback. To the best of our knowledge, this is the first theoretical confirmation that learning with bandit feedback is strictly harder than learning with full-information, even on a small finite action set and even in terms of the dependence on $T$ (previous gaps we are aware of were either in terms of the number of actions [4], or required large or continuous action spaces —see, e.g., [6, 21]). Moreover, recall the regret bound of $\mathcal{O}(\sqrt{T \log \log \log T})$ against the stochastic i.i.d. adversary with switching costs and bandit feedback. This demonstrates that dependencies in the loss process must play a crucial role in controlling the power of the switching costs adversary. Indeed, the $\Omega(T^{2/3})$ lower bound proven in the next section heavily relies on such dependencies.

Second, observe that in the full-information feedback case, the optimal regret against a switching costs adversary is $\Theta(\sqrt{T})$, whereas the optimal regret against the more general bounded memory adversary is $\Omega(T^{2/3})$. This is somewhat surprising given the ideas presented in [18] and later extended in [3]: The main technique used in these papers is to take an algorithm originally designed for oblivious adversaries, forcefully prevent it from switching actions very often, and obtain a new algorithm that guarantees a regret of $\mathcal{O}(T^{2/3})$ against bounded memory adversaries. This would

seem to imply that a small number of switches is the key to dealing with general bounded memory adversaries. Our result contradicts this intuition by showing that controlling the number of switches is easier then dealing with a general bounded memory adversary.

As noted above, our lower bounds require us to slightly weaken the standard technical assumption that loss values lie in a fixed interval $[0, C]$. We replace it with the following two assumptions:

1. *Bounded range*. We assume that the loss values *on each individual round* are bounded in an interval of constant size $C$, but we allow this interval to drift from round to round. Formally, $\forall t$, $\forall x_{1:t} \in \mathcal{A}^t$ and $\forall x'_{1:t} \in \mathcal{A}^t$,

$$\left| f_t(x_{1:t}) - f_t(x'_{1:t}) \right| \leq C . \tag{6}$$

2. *Bounded drift*. We also assume that the drift of each individual action from round to round is contained in a bounded interval of size $D_t$, where $D_t$ may grow slowly, as $\mathcal{O}\big(\sqrt{\log(t)}\big)$. Formally, $\forall t$ and $\forall x_{1:t} \in \mathcal{A}^t$,

$$\left| f_t(x_{1:t}) - f_{t+1}(x_{1:t}, x_t) \right| \leq D_t . \tag{7}$$

Since these assumptions are a relaxation of the standard assumption, all of the known lower bounds on regret automatically extend to our relaxed setting. For our results to be consistent with the current state of the art, we must also prove that all of the known upper bounds continue to hold after the relaxation, up to logarithmic factors.

## 2 Lower Bounds

In this section, we prove lower bounds on the player's expected regret in various settings.

### 2.1 $\Omega(T^{2/3})$ with Switching Costs and Bandit Feedback

We begin with a $\Omega(T^{2/3})$ regret lower bound against an oblivious adversary with switching costs, when the player receives bandit feedback. It is enough to consider a very simple setting, with only two actions, labeled 1 and 2. Using the notation introduced earlier, we use $\ell_1, \ell_2, \ldots$ to denote the oblivious sequence of loss functions chosen by the adversary before adding the switching cost.

**Theorem 1.** *For any player strategy that relies on bandit feedback and for any number of rounds $T$, there exist loss functions $f_1, \ldots, f_T$ that are oblivious with switching costs, with a range bounded by $C = 2$, and a drift bounded by $D_t = \sqrt{3 \log(t) + 16}$, such that $\mathbb{E}[R_T] \geq \frac{1}{40} T^{2/3}$.*

The full proof is given in the supplementary material, and here we give an informal proof sketch. We begin by constructing a *randomized* adversarial strategy, where the loss functions $\ell_1, \ldots, \ell_T$ are an instantiation of random variables $L_t, \ldots, L_T$ defined as follows. Let $\xi_1, \ldots, \xi_T$ be i.i.d. standard Gaussian random variables (with zero mean and unit variance) and let $Z$ be a random variable that equals $-1$ or $1$ with equal probability. Using these random variables, define for all $t = 1 \ldots T$

$$L_t(1) = \sum_{s=1}^{t} \xi_s , \qquad L_t(2) = L_t(1) + Z T^{-1/3} . \tag{8}$$

In words, $\{L_t(1)\}_{t=1}^{T}$ is simply a Gaussian random walk and $\{L_t(2)\}_{t=1}^{T}$ is the same random walk, slightly shifted up or down —see figure 1 for an illustration. It is straightforward to confirm that this loss sequence has a bounded range, as required by the theorem: by construction we have $|\ell_t(1) - \ell_t(2)| = T^{-1/3} \leq 1$ for all $t$, and since the switching cost can add at most 1 to the loss on each round, we conclude that $|f_t(1) - f_t(2)| \leq 2$ for all $t$. Next, we show that the expected regret of any player against this random loss sequence is $\Omega(T^{2/3})$, where expectation is taken over the randomization of both the adversary and the player. The intuition is that the player can only gain information about which action is better by switching between them. Otherwise, if he stays on the same action, he only observes a random walk, and gets no further information. Since the gap between the two losses on each round is $T^{-1/3}$, the player must perform $\Omega(T^{2/3})$ switches before he can identify the better action. If the player performs that many switches, the total regret incurred due to the switching costs is $\Omega(T^{2/3})$. Alternatively, if the player performs $o(T^{2/3})$ switches, he

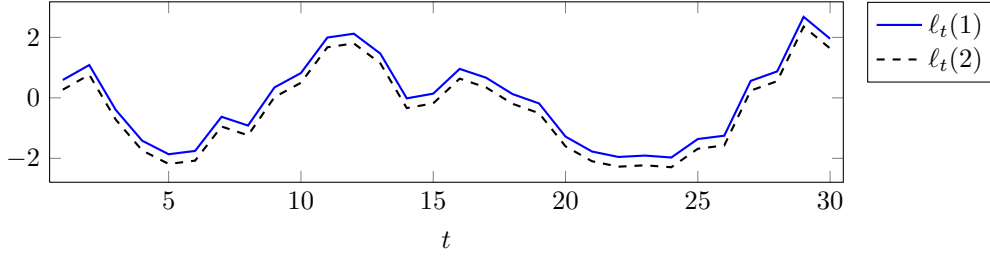

Figure 1: A particular realization of the random loss sequence defined in Eq. (8). The sequence of losses for action 1 follows a Gaussian random walk, whereas the sequence of losses for action 2 follows the same random walk, but slightly shifted either up or down.

can't identify the better action; as a result he suffers an expected regret of $\Omega(T^{-1/3})$ on each round and a total regret of $\Omega(T^{2/3})$.

Since the randomized loss sequence defined in Eq. (8), plus a switching cost, achieves an expected regret of $\Omega(T^{2/3})$, there must exist at least one *deterministic* loss sequence $\ell_1 \ldots \ell_T$ with a regret of $\Omega(T^{2/3})$. In our proof, we show that there exists such $\ell_1 \ldots \ell_T$ with bounded drift.

## 2.2 $\Omega(T^{2/3})$ with Bounded Memory and Full-Information Feedback

We build on Thm. 1 and prove a $\Omega(T^{2/3})$ regret lower bound in the *full-information setting*, where we get to see the entire loss vector on every round. To get this strong result, we need to give the adversary a little bit of extra power: memory of size 2 instead of size 1 as in the case of switching costs. To show this result, we again consider a simple setting with two actions.

**Theorem 2.** *For any player strategy that relies on full-information feedback and for any number of rounds $T \geq 2$, there exist loss functions $f_1, \ldots, f_T$, each with a memory of size $m = 2$, a range bounded by $C = 2$, and a drift bounded by $D_t = \sqrt{3 \log(t) + 18}$, such that $\mathbb{E}[R_T] \geq \frac{1}{40}(T-1)^{2/3}$.*

The formal proof is deferred to the supplementary material and a proof sketch is given here. The proof is based on a reduction from full-information to bandit feedback that might be of independent interest. We construct the adversarial loss sequence as follows: on each round, the adversary assigns the *same* loss to both actions. Namely, the value of the loss depends only on the player's previous two actions, and not on his action on the current round. Recall that even in the full-information version of the game, the player doesn't know what the losses would have been had he chosen different actions in the past. Therefore, we have made the full-information game as difficult as the bandit game. Specifically, we construct an oblivious loss sequence $\ell_1 \ldots \ell_T$ as in Thm. 1 and define

$$f_t(x_{1:t}) = \ell_{t-1}(x_{t-1}) + \mathbb{I}_{\{x_{t-1} \neq x_{t-2}\}} \ . \tag{9}$$

In words, we define the loss on round $t$ of the full-information game to be equal to the loss on round $t-1$ of a bandits-with-switching-costs game in which the player chooses the same sequence of actions. This can be done with a memory of size 2, since the loss in Eq. (9) is fully specified by the player's choices on rounds $t, t-1, t-2$. Therefore, the $\Omega(T^{2/3})$ lower bound for switching costs and bandit feedback extends to the full-information setting with a memory of size at least 2.

## 3 Upper Bounds

In this section, we show that the known upper bounds on regret, originally proved for bounded losses, can be extended to the case of losses with bounded range and bounded drift. Specifically, of the upper bounds that appear in Table 1, we prove the following:

- $\mathcal{O}(\sqrt{T})$ for an oblivious adversary with switching costs, with full-information feedback.
- $\widetilde{\mathcal{O}}(\sqrt{T})$ for an oblivious adversary with bandit feedback (where $\widetilde{\mathcal{O}}$ hides logarithmic factors).
- $\widetilde{\mathcal{O}}(T^{2/3})$ for a bounded memory adversary with bandit feedback.

The remaining upper bounds in Table 1 are either trivial or follow from the principle that an upper bound still holds if we weaken the adversary or provide a more informative feedback.

## 3.1 $\mathcal{O}(\sqrt{T})$ with Switching Costs and Full-Information Feedback

In this setting, $f_t(x_{1:t}) = \ell_t(x_t) + \mathbb{I}_{\{x_t \neq x_{t-1}\}}$. If the oblivious losses $\ell_1 \ldots \ell_T$ (without the additional switching costs) were all bounded in $[0, 1]$, the *Follow the Lazy Leader* (FLL) algorithm of [14] would guarantee a regret of $\mathcal{O}(\sqrt{T})$ with respect to these losses (again, without the additional switching costs). Additionally, FLL guarantees that its expected number of switches is $\mathcal{O}(\sqrt{T})$. We use a simple reduction to extend these guarantees to loss functions with a range bounded in an interval of size $C$ and with an arbitrary drift.

On round $t$, after choosing an action and receiving the loss function $\ell_t$, the player defines the modified loss $\ell_t'(x) = \frac{1}{C-1}\big(\ell_t(x) - \min_y \ell_t(y)\big)$ and feeds it to the FLL algorithm. The FLL algorithm then chooses the next action.

**Theorem 3.** *If each of the loss functions $f_1, f_2, \ldots$ is oblivious with switching costs and has a range bounded by $C$ then the player strategy described above attains $\mathcal{O}(C\sqrt{T})$ expected regret.*

The full proof is given in the supplementary material but the proof technique is straightforward. We first show that each $\ell_t'$ is bounded in $[0, 1]$ and therefore the standard regret bound for FLL holds with respect to the sequence of modified loss functions $\ell_1', \ell_2', \ldots$. Then we show that the guarantees provided for FLL imply a regret of $\mathcal{O}(\sqrt{T})$ with respect to the original loss sequence $f_1, f_2, \ldots$.

## 3.2 $\widetilde{\mathcal{O}}(\sqrt{T})$ with an Oblivious Adversary and Bandit Feedback

In this setting, $f_t(x_{1:t})$ simply equals $\ell_t(x_t)$. The reduction described in the previous subsection cannot be used in the bandit setting, since $\min_x \ell_t(x)$ is unknown to the player, and a different reduction is needed. The player sets a fixed horizon $T$ and focuses on controlling his regret at time $T$; he can then use a standard doubling trick [8] to handle an infinite horizon. The player uses the fact that each $f_t$ has a range bounded by $C$. Additionally, he defines $D = \max_{t \leq T} D_t$ and on each round he defines the modified loss

$$f_t'(x_{1:t}) = \frac{1}{2(C+D)}\big(\ell_t(x_t) - \ell_{t-1}(x_{t-1})\big) + \frac{1}{2}. \tag{10}$$

Note that $f_t'(X_{1:t})$ can be computed by the player using only bandit feedback. The player then feeds $f_t'(X_{1:t})$ to an algorithm that guarantees a $\mathcal{O}(\sqrt{T})$ *standard* regret (see definition in Eq. (2)) against a fixed action. The Exp3 algorithm, due to [4], is such an algorithm. The player chooses his actions according to the choices made by Exp3. The following theorem states that this reduction results in a bandit algorithm that guarantees a regret of $\widetilde{\mathcal{O}}(\sqrt{T})$ against oblivious adversaries.

**Theorem 4.** *If each of the loss functions $f_1 \ldots f_T$ is oblivious with a range bounded by $C$ and a drift bounded by $D_t = \mathcal{O}\big(\sqrt{\log(t)}\big)$ then the player strategy described above attains $\widetilde{\mathcal{O}}(C\sqrt{T})$ expected regret.*

The full proof is given in the supplementary material. In a nutshell, we show that each $f_t'$ is a loss function bounded in $[0, 1]$ and that the analysis of Exp3 guarantees a regret of $\mathcal{O}(\sqrt{T})$ with respect to the loss sequence $f_1' \ldots f_T'$. Then, we show that this guarantee implies a regret of $(C+D)\mathcal{O}(\sqrt{T}) = \widetilde{\mathcal{O}}(C\sqrt{T})$ with respect to the original loss sequence $f_1 \ldots f_T$.

## 3.3 $\widetilde{\mathcal{O}}(T^{2/3})$ with Bounded Memory and Bandit Feedback

Proving an upper bound against an adversary with a memory of size $m$, with bandit feedback, requires a more delicate reduction. As in the previous section, we assume a finite horizon $T$ and we let $D = \max_t D_t$. Let $K = |\mathcal{A}|$ be the number of actions available to the player.

Since $f_T(x_{1:t})$ depends only on the last $m+1$ actions in $x_{1:t}$, we slightly overload our notation and define $f_t(x_{t-m:t})$ to mean the same as $f_t(x_{1:t})$. To define the reduction, the player fixes a base

action $x_0 \in \mathcal{A}$ and for each $t > m$ he defines the loss function

$$\widehat{f}_t(x_{t-m:t}) = \frac{1}{2(C + (m+1)D)}\big(f_t(x_{t-m:t}) - f_{t-m-1}(x_0 \ldots x_0)\big) + \frac{1}{2} \ .$$

Next, he divides the $T$ rounds into $J$ consecutive epochs of equal length, where $J = \Theta(T^{2/3})$. We assume that the epoch length $T/J$ is at least $2K(m+1)$, which is true when $T$ is sufficiently large. At the beginning of each epoch, the player plans his action sequence for the entire epoch. He uses some of the rounds in the epoch for exploration and the rest for exploitation. For each action in $\mathcal{A}$, the player chooses an exploration interval of $2(m+1)$ consecutive rounds within the epoch. These $K$ intervals are chosen randomly, but they are not allowed to overlap, giving a total of $2K(m+1)$ exploration rounds in the epoch. The details of how these intervals are drawn appears in our analysis, in the supplementary material. The remaining $T/J - 2K(m+1)$ rounds are used for exploitation.

The player runs the Hedge algorithm [11] in the background, invoking it only at the beginning of each epoch and using it to choose one exploitation action that will be played consistently on all of the exploitation rounds in the epoch. In the exploration interval for action $x$, the player first plays $m+1$ rounds of the base action $x_0$ followed by $m+1$ rounds of the action $x$. Letting $t_x$ denote the first round in this interval, the player uses the observed losses $f_{t_x+m}(x_0 \ldots x_0)$ and $f_{t_x+2m+1}(x \ldots x)$ to compute $\widehat{f}_{t_x+2m+1}(x \ldots x)$. In our analysis, we show that the latter is an unbiased estimate of the average value of $\widehat{f}_t(x \ldots x)$ over $t$ in the epoch. At the end of the epoch, the $K$ estimates are fed as feedback to the Hedge algorithm.

We prove the following regret bound, with the proof deferred to the supplementary material.

**Theorem 5.** *If each of the loss functions $f_1 \ldots f_T$ is has a memory of size $m$, a range bounded by $C$, and a drift bounded by $D_t = \mathcal{O}\big(\sqrt{\log(t)}\big)$ then the player strategy described above attains $\widetilde{\mathcal{O}}(T^{2/3})$ expected regret.*

## 4 Discussion

In this paper, we studied the problem of prediction with expert advice against different types of adversaries, ranging from the oblivious adversary to the general adaptive adversary. We proved upper and lower bounds on the player's regret against each of these adversary types, in both the full-information and the bandit feedback models. Our lower bounds essentially matched our upper bounds in all but one case: the adaptive adversary with a unit memory in the full-information setting, where we only know that regret is $\Omega(\sqrt{T})$ and $\mathcal{O}(T^{2/3})$. Our bounds have two important consequences. First, we characterize the regret attainable with switching costs, and show a setting where predicting with bandit feedback is strictly more difficult than predicting with full-information feedback —even in terms of the dependence on $T$, and even on small finite action sets. Second, in the full-information setting, we show that predicting against a switching costs adversary is strictly easier than predicting against an arbitrary adversary with a bounded memory. To obtain our results, we had to relax the standard assumption that loss values are bounded in $[0, 1]$. Re-introducing this assumption and proving similar lower bounds remains an elusive open problem. Many other questions remain unanswered. Can we characterize the dependence of the regret on the number of actions? Can we prove regret bounds that hold with high probability? Can our results be generalized to more sophisticated notions of regret, as in [3]?

In addition to the adversaries discussed in this paper, there are other interesting classes of adversaries that lie between the oblivious and the adaptive. A notable example is the family of *deterministically adaptive* adversaries, which includes adversaries that adapt to the player's actions in a known deterministic way, rather than in a secret malicious way. For example, imagine playing a multi-armed bandit game where the loss values are initially oblivious, but whenever the player chooses an arm with zero loss, the loss of the same arm on the next round is deterministically changed to zero. Many real-world online prediction scenarios are deterministically adaptive, but we lack a characterization of the expected regret in this setting.

**Acknowledgments**

Part of this work was done while NCB was visiting OD at Microsoft Research, whose support is gratefully acknowledged.

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
