[Supplementary Material]

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

# Appendices

## A  Distribution-free regret bound for bandits with switching costs

In this appendix we adapt results of [7] to show a strategy that achieves $\mathcal{O}\big(\sqrt{T \log\log\log T}\big)$ regret against any i.i.d. oblivious adversary in the bandit setting with switching costs, assuming a finite action set $\mathcal{A} = \{1 \dots K\}$. The strategy used by this stochastic adversary is specified by a probability distribution over oblivious loss functions. The oblivious loss function for each step $t = 1, 2, \dots$ is the realization on an independent draw $L_t$ from this distribution. The regret of a player choosing actions $X_0 = X_1, X_2, \dots$ is defined by

$$R_T \;=\; \sum_{t=1}^{T} \mathbb{E}_t\big[L_t(X_t) + \mathbb{I}_{\{X_t \neq X_{t-1}\}}\big] - \min_{x \in \mathcal{A}} \sum_{t=1}^{T} \mathbb{E}\big[L_t(x)\big]$$

where the expectation $\mathbb{E}$ is over the random draw of each $L_t$ and the possible randomization of the player, and the expectation $\mathbb{E}_t$ is conditioned over $X_1, L_1(X_1), \dots, X_{t-1}, L_{t-1}(X_{t-1})$.

Our result focuses on loss distributions such that the law of each marginal $L_1(x)$ is subgaussian. A random variable $Z$ is subgaussian if there exist constants $b, c$ such that for any $a > 0$ $\mathbb{P}\big(Z > \mathbb{E}\, Z + a\big) \leq b e^{-ca^2}$ and $\mathbb{P}\big(Z < \mathbb{E}\, Z - a\big) \leq b e^{-ca^2}$. One can then show that, for any i.i.d. sequence $Z_1, \dots, Z_T$ of subgaussian random variables,

$$\mathbb{P}\left( \left| \frac{1}{T} \sum_{t=1}^{T} Z_t - \mathbb{E}\, Z_1 \right| > \sqrt{\frac{112b}{cT} \ln \frac{1}{\delta}} \right) \leq \delta \;. \tag{11}$$

In the following, we use the notation $\mathbb{E}\big[L_t(x)\big] = \mu(x)$ and $\mu^* = \min_{x \in \mathcal{A}} \mu(x)$ .

**Theorem 6.** *Consider a finite action set $\mathcal{A} = \{1 \dots K\}$. Then for each $T$ there exists a deterministic player strategy for the bandit game with i.i.d. oblivious adversaries and switching costs, whose regret after $T$ steps is $\mathcal{O}\big(\sqrt{T \log\log\log T}\big)$ with high probability, provided the distribution of $L_1(x)$ is sugaussian for each $x \in \mathcal{A}$.*

*Proof.* Consider the following player that proceeds in stages. At each stage $s = 1, 2, \dots, S$, the player maintains a set $A_s \subseteq \mathcal{A}$ of active actions. Each action is played $T_s / |A_s|$ times in a round-robin fashion, where $T_s = T^{1-2^{-s}}$ is the total number of plays in stage $s$ and $T$ is the known horizon. Note that the overall number of switches is at most $KS$, where

$$S = \min \left\{ j \in \mathbb{N} \;:\; \sum_{s=1}^{j} T_s \geq T \right\} = \mathcal{O}\big(\ln\ln T\big) \;.$$

Let $\widehat{\mu}_s(x)$ the sample mean of losses for action $x$ in stage $s$, and define

$$\widehat{x}_s = \operatorname*{argmin}_{x \in A_s} \widehat{\mu}_s(x)$$

the best empirical action in stage $s$. The sets $A_s$ of active actions are defined as follows: $A_1 = \mathcal{A}$ and

$$A_s = \left\{ x \in A_{i-1} \;:\; \widehat{\mu}_{s-1}(x) \leq \widehat{\mu}_{s-1}(\widehat{x}_{s-1}) + 2C_{s-1} \right\}$$

where

$$C_s = \sqrt{112(b/c) \frac{K}{T_s} \ln \frac{KS}{\delta}} \;.$$

Note that $A_S \subseteq \cdots \subseteq A_1$ by construction. Also, using (11) and the union bound we have that

$$\max_{x \in A_s} \big| \widehat{\mu}_s(x) - \mu(x) \big| \leq C_s \tag{12}$$

simultaneously for all $s = 1, \dots, S$ with probability at least $1 - \delta$.

We claim the following.

**Claim 1.** *With probability at least $1 - \delta$,*

$$x^* \in \bigcap_{s=1}^{S} A_s \qquad \text{and} \qquad 0 \leq \widehat{\mu}_s(x^*) - \widehat{\mu}_s(\widehat{x}_s) \leq 2C_s \quad \text{for all } s = 1, \dots, S.$$

*Proof of Claim.* We prove the lemma by induction on $s = 1, \dots, S$. We first show that the base case $s = 1$ holds with probability at least $1 - \delta/S$. Then we show that if the claim holds for $s - 1$, then it holds for $s$ with probability at least $1 - \delta/S$ over all random events in stage $s$. Therefore, using a union bound over $s = 1, \dots, S$ we get that the claim holds simultaneously for all $s$ with probability at least $1 - \delta$.

For the base case $s = 1$ note that $x^* \in A_1$ by definition, and thus $\widehat{\mu}_1(\widehat{x}_1) \leq \widehat{\mu}_1(x^*)$ holds. Moreover, using (12) we obtain that

$$\widehat{\mu}_1(x^*) - \mu(x^*) \leq C_1 \qquad \text{and} \qquad \mu(\widehat{x}_1) - \widehat{\mu}_1(\widehat{x}_1) \leq C_1$$

holds with probability at least $1 - \delta/S$. Since $\mu(x^*) - \mu(\widehat{x}_1) \leq 0$ by definition of $x^*$, we obtain

$$0 \leq \widehat{\mu}_1(x^*) - \widehat{\mu}_1(\widehat{x}_1) \leq 2C_1$$

as required. We now prove the claim for $s > 1$ using the inductive assumption

$$x^* \in A_{s-1} \qquad \text{and} \qquad 0 \leq \widehat{\mu}_{s-1}(x^*) - \widehat{\mu}_{s-1}(\widehat{x}_{s-1}) \leq 2C_{s-1} .$$

The inductive assumption directly implies that $x^* \in A_s$. Thus we have $\widehat{\mu}_i(\widehat{x}_s) \leq \widehat{\mu}_s(x^*)$, because $\widehat{x}_s$ minimizes $\widehat{\mu}_s$ over a set that contains $x^*$. The rest of the proof of the claim closely follows that of the base case $s = 1$. $\square$

Now, for any $s = 1, \dots, S$ and for any $x \in A_s$ we have that

$$
\begin{aligned}
\mu(x) - \mu(x^*) &\leq \widehat{\mu}_{s-1}(x) - \mu(x^*) + C_{s-1} && \text{by (12)} \\
&\leq \widehat{\mu}_{s-1}(\widehat{x}_{s-1}) - \mu(x^*) + 3C_{s-1} && \text{by definition of } A_{s-1}, \text{ since } x \in A_s \subseteq A_{s-1} \\
&\leq \widehat{\mu}_{s-1}(x^*) - \mu(x^*) + 3C_{s-1} && \text{since } \widehat{x}_{s-1} \text{ minimizes } \widehat{\mu}_{s-1} \text{ in } A_{s-1} \\
&\leq 4C_{s-1} && \text{by (12)}
\end{aligned}
$$

holds with probability at least $1 - \delta/S$. Hence, recalling that

$$\sum_{t=1}^{T} \mathbb{I}_{\{X_t \neq X_{t-1}\}} \leq KS$$

holds deterministically, the regret of the player over the $T$ plays can be bounded as follows

$$
\begin{aligned}
KS + \sum_{t=1}^{T} \big(\mu(X_t) - \mu^*\big) &= KS + \sum_{s=1}^{S} \frac{T_s}{|A_s|} \sum_{x \in A_s} \big(\mu(x) - \mu^*\big) \\
&= KS + \frac{T_1}{K} \sum_{i=1}^{K} \big(\mu(x) - \mu^*\big) + \sum_{s=2}^{S} \frac{T_s}{|A_s|} \sum_{x \in A_s} \big(\mu(x) - \mu^*\big) \\
&\leq KS + T_1 \mu^* + \sum_{i=2}^{S} 4 T_s \sqrt{112(b/c) \frac{K}{T_s} \ln \frac{KS}{\delta}} \\
&= KS + T_1 \mu^* + 4 \sqrt{112(b/c) K \ln \frac{KS}{\delta}} \sum_{s=2}^{S} \frac{T_s}{\sqrt{T_{s-1}}}
\end{aligned}
$$

Now, since $T_1 = \sqrt{T}$, $T_s/\sqrt{T_{s-1}} = \sqrt{T}$ and $S = \mathcal{O}(\ln \ln T)$, we obtain that with probability at least $1 - \delta$ the regret is at most of order

$$K \ln \ln T + \mu^* \sqrt{T} + \sqrt{KT \left( \ln \frac{K}{\delta} + \ln \ln \ln T \right)}$$

as desired. $\square$

# B  Proof of Thm. 1

As mentioned in the text, we first consider the player's expected regret against a randomized adversary. Specifically, we define

$$\forall t \quad L_t(1) = \sum_{s=1}^{t} \xi_s \quad \text{and} \quad L_t(2) = L_t(1) + Z\epsilon \ ,$$

where $\xi_1 \ldots \xi_T$ are independent standard Gaussians, $Z$ equals $-1$ or $1$ with equal probability, and $\epsilon$ is the gap between the losses of the two actions (which will later be set to $\epsilon = T^{-1/3}$).

Next, we assume for now, without loss of generality, that the player is deterministic. A deterministic player chooses each action $X_t$ as a deterministic function of the random losses suffered on the previous rounds, $L_1(X_1) \ldots L_{t-1}(X_{t-1})$. We can make this assumption because any randomized player strategy can be seen as a distribution over deterministic player strategies, and since the randomization used by the adversary is independent of the player's strategy.

In the results below, $\mathbb{P}$ denotes the distribution of the randomized adversary. We also introduce the conditional distributions $\mathbb{S} = \mathbb{P}(\cdot \mid Z > 0)$ (i.e., 1 is the better action) and $\mathbb{Q} = \mathbb{P}(\cdot \mid Z < 0)$ (i.e., 2 is the better action). Since $Z$ has an equal probability of being negative or positive, it holds that $\mathbb{P} = \frac{1}{2}(\mathbb{S} + \mathbb{Q})$.

We begin with the following technical lemma.

**Lemma 1.** *Let* $\mathbb{I}_{\{x_{t-1} \neq x_t\}}$ *indicate whether the player switched actions on round $t$ (and $1$ for $t = 1$). Then for any event $A$,*

$$\left| \mathbb{S}(A) - \mathbb{Q}(A) \right| \leq \epsilon \sqrt{\mathbb{E}\left[ \sum_{t=1}^{T} \mathbb{I}_{\{X_t \neq X_{t-1}\}} \right]}$$

*where the expectation in the right-hand side is with respect to $\mathbb{P}$.*

*Proof.* To show this, we use the chain rule for relative entropy, which implies

$$D_{KL}\left(\mathbb{S} \,\|\, \mathbb{Q}\right) = \sum_{t=1}^{T} D_{KL}\left(\mathbb{S}_{t-1} \,\|\, \mathbb{Q}_{t-1}\right) \tag{13}$$

where $\mathbb{S}_{t-1}$ and $\mathbb{Q}_{t-1}$ denote the distributions of the player's loss $L_t(x_t)$ conditioned on $L_1, \ldots, L_{t-1}$, when the joint distribution of $L_1, \ldots, L_T$ is, respectively, $\mathbb{S}$ and $\mathbb{Q}$.

Let us focus on a particular term $D_{KL}\left(\mathbb{S}_{t-1} \,\|\, \mathbb{Q}_{t-1}\right)$ and a particular realization of the random losses $L_1, \ldots, L_{t-1}$. Since we assume a deterministic player strategy, for any such realization the player's choices $x_{1:t}$ are all determined, and we deterministically have that the player either switched or not at time $t$. If he did not switch, then $L_t(x_t)$ is distributed as $L_{t-1}(x_{t-1}) + \xi_t$ under both measures $\mathbb{S}_{t-1}$ and $\mathbb{Q}_{t-1}$, so the relative entropy between them is zero. If he did switch, then $L_t(x_t)$ is distributed as $L_{t-1}(x_{t-1}) - \epsilon + \xi$ under $\mathbb{S}_{t-1}$ (where the switch is towards the best action), and as $L_{t-1}(x_{t-1}) + \epsilon + \xi$ under $\mathbb{Q}_{t-1}$ (where the switch is towards the worst action). Hence, the relative entropy is the same as two standard Gaussians whose means are shifted by $2\epsilon$, namely $2\epsilon^2$. So overall, we can upper bound Eq. (13) by

$$2\epsilon^2 \, \mathbb{E}\left[ \sum_{t=1}^{T} \mathbb{I}_{\{X_t \neq X_{t-1}\}} \,\middle|\, Z > 0 \right] . \tag{14}$$

Using a similar argument, we also show that $D_{KL}\left(\mathbb{Q} \,\|\, \mathbb{S}\right)$ is upper bounded by Eq. (14) in which the conditioning on $Z > 0$ is replaced by $Z < 0$. Then, Pinsker's inequality implies that $\left| \mathbb{S}(A) - \mathbb{Q}(A) \right|^2$ is at most

$$\frac{\epsilon^2}{2} \left( \mathbb{E}\left[ \sum_{t=1}^{T} \mathbb{I}_{\{X_t \neq X_{t-1}\}} \,\middle|\, Z > 0 \right] + \mathbb{E}\left[ \sum_{t=1}^{T} \mathbb{I}_{\{X_t \neq X_{t-1}\}} \,\middle|\, Z < 0 \right] \right) = \epsilon^2 \mathbb{E}\left[ \sum_{t=1}^{T} \mathbb{I}_{\{X_t \neq X_{t-1}\}} \right]$$

which gives the desired bound. $\square$

With this lemma, we can prove a lower bound on the expected regret for randomized adversaries.

**Lemma 2.** *By picking $\epsilon = T^{-1/3}$, the expected regret of any deterministic player strategy, over the randomness of the adversary, is at least $\frac{1}{10}T^{2/3}$.*

*Proof.* Let $A$ be the event that the worst action (action 2 if $Z > 0$, and 1 if $Z < 0$) was picked by the player at least $T/2$ times. Also, let $S_T = \sum_{t=1}^{T} \mathbb{I}_{\{X_t \neq X_{t-1}\}}$ be the number of switches the player performs. Then

$$\mathbb{E}[R_T] \geq \mathbb{E}\left[\max\left\{S_T, \frac{\epsilon T}{2}\mathbb{I}_{\{A\}}\right\}\right] \geq \mathbb{E}\left[\frac{1}{2}\left(S_T + \frac{\epsilon T}{2}\mathbb{I}_{\{A\}}\right)\right] = \frac{1}{2}\mathbb{E}[S_T] + \frac{\epsilon T}{4}\mathbb{P}(A).$$

Moreover, letting $A_1$ denote the event that the player chose action 1 at least $T/2$ times, and letting $A_2$ denote the event that the player chose action 2 at least $T/2$ times, we have $\mathbb{P}(A) = \frac{1}{2}\big(\mathbb{S}(A_2) + \mathbb{Q}(A_1)\big)$. Substituting this, we get

$$\frac{1}{2}\mathbb{E}[S_T] + \frac{\epsilon T}{8}\big(\mathbb{S}(A_2) + \mathbb{Q}(A_1)\big).$$

Using Lemma 1 to lower bound $\mathbb{Q}(A_1)$ via $\mathbb{S}(A_1)$, we get a lower bound of

$$\frac{1}{2}\mathbb{E}[S_T] + \frac{\epsilon T}{8}\left(\mathbb{S}(A_2) + \mathbb{S}(A_1) - \epsilon\sqrt{\mathbb{E}[S_T]}\right) \geq \frac{1}{2}\mathbb{E}[S_T] + \frac{\epsilon T}{8}\left(\mathbb{S}(A_1 \cup A_2) - \epsilon\sqrt{\mathbb{E}[S_T]}\right)$$

$$= \frac{1}{2}\mathbb{E}[S_T] + \frac{\epsilon T}{8}\left(1 - \epsilon\sqrt{\mathbb{E}[S_T]}\right) = \frac{1}{2}\mathbb{E}[S_T] - \frac{\epsilon^2 T}{8}\sqrt{\mathbb{E}[S_T]} + \frac{\epsilon T}{8},$$

where we used a union bound and the fact that either $A_1$ or $A_2$ always holds. This is a quadratic function of $\sqrt{\mathbb{E}[S_T]}$, and it is easily verified that the lowest possible value it can attain (for any value of $\mathbb{E}[S_T]$) is

$$\frac{\epsilon T}{8} - \frac{\epsilon^4 T^2}{128}.$$

Picking $\epsilon = T^{-1/3}$, this equals $\left(\frac{1}{8} - \frac{1}{128}\right)T^{2/3} > \frac{1}{10}T^{2/3}$. $\qquad\square$

The lemma above tells us that for the randomized adversary strategy we have devised, the expected regret for any deterministic player is at least $\frac{1}{10}T^{2/3}$. This implies that there exist some *deterministic* adversarial strategy, for which the expected regret of any *possibly randomized* player is at least $\frac{1}{10}T^{2/3}$. However, we are not done yet, since this strategy doesn't guarantee that the losses have bounded drift: In our case, the variation is governed by a potentially unbounded Gaussian random variable, so the deterministic adversary strategy that we picked might have an arbitrarily large drift. So now, our goal will be to show that there exists some deterministic adversarial strategy for which the expected regret is large, *and the variation is bounded*. To do this, the plan is to show that the probabilities (over the adversary's strategy) of the two events are large, summing to a number larger than one. This means there is some realization of the losses such that both events occur. We first state and prove two auxiliary lemmas, and then provide two more fundamental lemmas which together give us the required result.

**Lemma 3.** *Let $Y$ be a random variable in $[-b, b]$ (where $b > 0$), and $\mathbb{E}[Y] \geq c$ for some $c \in [0, b/2]$. Then we have*

$$\mathbb{P}(Y \geq c/2) \geq \frac{c}{2b - c} \geq \frac{c}{2b}.$$

*Proof.*

$$c \leq \mathbb{E}[Y] = \mathbb{P}(Y \geq c/2)\mathbb{E}[Y \mid Y \geq c/2] + \mathbb{P}(Y < c/2)\mathbb{E}[Y \mid Y < c/2]$$
$$\leq \mathbb{P}(Y \geq c/2)b + \big(1 - \mathbb{P}(Y \geq c/2)\big)c/2$$

Solving for $\mathbb{P}(Y \geq c/2)$ gives the desired result. $\qquad\square$

**Lemma 4.** *Let $\xi_1, \xi_2, \ldots$ be an infinite sequence of independent standard Gaussian random variables. Then for any $\delta \in (0, 1)$*

$$\mathbb{P}\left(\exists t \ : \ |\xi_t| \geq \sqrt{3\log(2t/\delta)}\right) \leq \delta.$$

*Proof.* By a standard Gaussian tail bound, we have that $\mathbb{P}(|\xi_t| > x) \leq \exp(-x^2/2)$ for any $x \geq 0$. This implies that

$$\mathbb{P}(|\xi_t| \geq \sqrt{3 \log(2t/\delta)}) \leq \left(\frac{\delta}{2t}\right)^{3/2}.$$

By a union bound, we get that

$$\mathbb{P}\left(\exists t \ : \ |\xi_t| \geq \sqrt{3 \log(2t/\delta)}\right) \leq \sum_{t=1}^{\infty} \left(\frac{\delta}{2t}\right)^{3/2} \leq \delta^{3/2} < \delta.$$

$\square$

**Lemma 5.** *For any (possibly randomized) player strategy, it holds that*

$$\mathbb{P}\left(\mathbb{E}_{player}[R_T] \geq \frac{1}{40} T^{2/3}\right) \geq \frac{1}{40},$$

*where $\mathbb{P}$ is over the adversary's randomization, and $\mathbb{E}_{player}[R_T]$ is the player's expected regret (over the player's randomization).*

*Proof.* By Lemma 2, we already know that

$$\mathbb{E}\left[\mathbb{E}_{\text{player}}[R_T]\right] \geq \frac{1}{10} T^{2/3}, \tag{15}$$

since if we have a $T^{2/3}/10$ lower bound on the regret for any deterministic player strategy, the same holds for any randomized player strategy. Our approach is to apply Lemma 3 in order to convert this into a probability lower bound as in the lemma statement. However, we cannot apply Lemma 3 as-is, since $\mathbb{E}_{\text{player}}[R_T]$ can be as large as $\Omega(T)$, and the resulting bound is too weak. Instead, we show that there exists a *different* player strategy, with expected regret $\mathbb{E}_{\widetilde{\text{player}}}[R_T]$, such that $|\mathbb{E}_{\widetilde{\text{player}}}[R_T]|$ is always at most $2T^{2/3}$ and

$$\mathbb{E}_{\widetilde{\text{player}}}[R_T] \leq 2 \, \mathbb{E}_{\text{player}}[R_T] \tag{16}$$

for any realization of the adversary's random strategy. Also, analogous to Eq. (15), we have $\mathbb{E}[\mathbb{E}_{\widetilde{\text{player}}}[R_T]] \geq \frac{1}{10} T^{2/3}$ by Lemma 2. Therefore, using Eq. (16) and Lemma 3, we get that

$$\mathbb{P}\left(\mathbb{E}_{\text{player}}[R_T] \geq \frac{1}{40} T^{2/3}\right) \geq \mathbb{P}\left(\mathbb{E}_{\widetilde{\text{player}}}[R_T] \geq \frac{1}{20} T^{2/3}\right) \geq \frac{1}{40}$$

as required.

The new player strategy we consider depends on the horizon $T$, and is very simple: It is identical to the original player strategy, but whenever the number of action switches reaches $\lfloor T^{2/3} \rfloor$, the player "freezes" in its current action, and keeps playing the same action till $T$ rounds are elapsed. Clearly, the number of switches with this strategy can never be more than $T^{2/3}$, and since the regret in terms of the loss $\ell_t$ at each round is either $0$ or $T^{-1/3}$, we get that the total regret $R_T$ can never be more than $T^{2/3} + T * T^{-1/3} = 2T^{2/3}$.

To prove Eq. (16), we consider some instantiation of the adversary's random strategy, and note that for any realization of the player's random coin tosses, the regret can only differ between the two strategies if $S_T$ (the total number of switches) is at least $\lfloor T^{2/3} \rfloor$. Therefore, we have $\mathbb{P}_{\text{player}}\left(S_T < \lfloor T^{2/3} \rfloor\right) = \mathbb{P}_{\widetilde{\text{player}}}\left(S_T < \lfloor T^{2/3} \rfloor\right)$, $\mathbb{P}_{\text{player}}\left(S_T \geq \lfloor T^{2/3} \rfloor\right) = \mathbb{P}_{\widetilde{\text{player}}}\left(S_T \geq \lfloor T^{2/3} \rfloor\right)$ and $\mathbb{E}_{\text{player}}[R_T | S_T < \lfloor T^{2/3} \rfloor] = \mathbb{E}_{\widetilde{\text{player}}}[R_T | S_T < \lfloor T^{2/3} \rfloor]$. Also, we recall that $R_T \geq 0$ with the adversary strategy that we consider (since one action is always worse than the other action at all rounds). Finally, we note that if $S_T \geq \lfloor T^{2/3} \rfloor$, then the regret for both strategies is at least $\lfloor T^{2/3} \rfloor$ (since with the adversary strategy that we consider, the number of switches is a lower bound on the

regret). Using these observations, we have

$$\mathbb{E}_{\widetilde{\text{player}}}[R_T]$$

$$= \mathbb{P}_{\widetilde{\text{player}}}(S_T < \lfloor T^{2/3} \rfloor)\mathbb{E}_{\widetilde{\text{player}}}[R_T | S_T < \lfloor T^{2/3} \rfloor] + \mathbb{P}_{\widetilde{\text{player}}}(S_T \geq \lfloor T^{2/3} \rfloor)\mathbb{E}_{\widetilde{\text{player}}}[R_T | S_T \geq \lfloor T^{2/3} \rfloor]$$

$$\leq \mathbb{P}_{\widetilde{\text{player}}}(S_T < \lfloor T^{2/3} \rfloor)\mathbb{E}_{\widetilde{\text{player}}}[R_T | S_T < \lfloor T^{2/3} \rfloor] + \mathbb{P}_{\widetilde{\text{player}}}(S_T \geq \lfloor T^{2/3} \rfloor)2T^{2/3}$$

$$= \mathbb{P}_{\text{player}}(S_T < \lfloor T^{2/3} \rfloor)\mathbb{E}_{\text{player}}[R_T | S_T < \lfloor T^{2/3} \rfloor] + \mathbb{P}_{\text{player}}(S_T \geq \lfloor T^{2/3} \rfloor)2T^{2/3}$$

$$\leq 2\left(\mathbb{P}_{\text{player}}(S_T < \lfloor T^{2/3} \rfloor)\mathbb{E}_{\text{player}}[R_T | S_T < \lfloor T^{2/3} \rfloor] + \mathbb{P}_{\text{player}}(S_T \geq \lfloor T^{2/3} \rfloor)T^{2/3}\right)$$

$$\leq 2\left(\mathbb{P}_{\text{player}}(S_T < \lfloor T^{2/3} \rfloor)\mathbb{E}_{\text{player}}[R_T | S_T < \lfloor T^{2/3} \rfloor] + \mathbb{P}_{\text{player}}(S_T \geq \lfloor T^{2/3} \rfloor)\mathbb{E}_{\text{player}}[R_T | S_T \geq \lfloor T^{2/3} \rfloor]\right)$$

$$= 2\,\mathbb{E}_{\text{player}}[R_T],$$

where in the second-to-last step we used the fact that if $S_T \geq \lfloor T^{2/3} \rfloor$, then the regret is at least $\lfloor T^{2/3} \rfloor$, plus we must have picked the worst action (worst by $T^{-1/3}$ than the best action) at least $\Omega(T^{2/3})$ times, hence the total regret is certainly at least $T^{2/3}$. $\qquad\square$

Finally, we use Lemma 4 with $\delta = 1/80$, to get that with probability at least $1 - 1/80$, the drift factor $D_t$ of the adversarial strategy is at most $\sqrt{3\log(160t)} \leq \sqrt{3\log(t) + 16}$ for all $t$. Moreover, Lemma 5 tells us that $\mathbb{E}_{\text{player}}[R_T]$ is at least $\frac{1}{40}T^{2/3}$ with probability at least $1/40$. This implies that the intersection of the two events is non-empty, and there exists some deterministic adversarial strategy, such that the drift $D_t \leq \sqrt{3\log(t) + 16}$ for all $t$, *and* the expected regret is at least $\frac{1}{40}T^{2/3}$ as required.

## C  Proof of Thm. 2

Thm. 1 guarantees that given any player's strategy, there is some deterministic adversary strategy with a lower bound on the regret. However, as part of proving Thm. 1, we actually showed that there exists some *randomized* adversary strategy $\{\hat{f}_t\}_{t=1}^T$ with memory size 1, such that for *any* (possibly randomized) player strategy $x_{1:t}$,

$$\mathbb{E}\left[\sum_{t=1}^T \hat{f}_t(X_{t-1}, X_t) - \min_{x \in \mathcal{A}} \sum_{t=1}^T \hat{f}_t(x, x)\right] \geq \frac{1}{10}T^{2/3} \tag{17}$$

(see Lemma 2). We now use this strategy to define a randomized adversary strategy for our setting (with memory size 2), for a game of $T + 1$ rounds. We let $f_1(x_1) = 0$ for any $x_1$, $f_2(x_1, x_2) = \hat{f}_1(x_1)$, and for every $t = 3 \ldots T + 1$,

$$f_t(x_{t-2}, x_{t-1}, x_t) = \hat{f}_{t-1}(x_{t-2}, x_{t-1}) . \tag{18}$$

Now, suppose we had some (possibly randomized) player strategy $X_1 \ldots X_{T+1}$, so that in expectation over the player and adversary strategies, we have

$$\mathbb{E}\left[\sum_{t=1}^{T+1} f_t(X_{t-2}, X_{t-1}, X_t) - \min_{x \in \mathcal{A}} \sum_{t=1}^{T+1} f_t(x, x, x)\right] < \frac{1}{10}T^{2/3}.$$

In particular, since $f_1$ is always 0, it would imply that

$$\mathbb{E}\left[\sum_{t=2}^{T+1} f_t(X_{t-2}, X_{t-1}, X_t) - \min_{x \in \mathcal{A}} \sum_{t=2}^{T+1} f_t(x, x, x)\right] < \frac{1}{10}T^{2/3} .$$

By Eq. (18), this implies

$$\mathbb{E}\left[\sum_{t=1}^T \hat{f}_t(X_{t-1}, X_t) - \min_{x \in \mathcal{A}} \sum_{t=1}^T \hat{f}_t(x, x)\right] < \frac{1}{10}T^{2/3} .$$

Thus, if we could implement the player strategy $X_1 \ldots X_T$ in the bandits-with-switching-costs setting, it will contradict Eq. (17). To see that this indeed can happen, note that each $X_t$ is a (possibly

randomized) function of $X_{1:t-1}$ as well as $\{f_\tau(X_{\tau-2}, X_{\tau-1}, X_\tau)\}_{\tau=1}^{t-1}$. But again, due to Eq. (18) and the fact that $f_1$ is always 0, $X_t$ can in fact be defined using $X_{1:t-1}$ and

$$\left\{f_\tau(X_{\tau-2}, X_{\tau-1}, X_\tau)\right\}_{\tau=2}^{t-1} \;=\; \left\{\hat{f}_{\tau-1}(X_{\tau-2}, X_{\tau-1})\right\}_{\tau=2}^{t-1} \,.$$

The right hand side is an observable quantity in the bandit setting: In each round $t$, we know what are the set of losses $\{\hat{f}_{\tau-1}(X_{\tau-2}, X_{\tau-1})\}_{\tau=2}^{t-1}$ that we obtained. Thus, we can simulate the strategy $x_{1:t}$ in the bandit-with-switching-costs setting, and get an expected regret smaller than $\frac{1}{10}T^{2/3}$, contradicting Eq. (17). Thus, the expected regret (for a game of $T+1$ rounds) must be at least $\frac{1}{10}T^{2/3}$. Substituting $T$ instead of $T+1$, we get that the expected regret for a game with $T$ rounds is at least $\frac{1}{10}(T-1)^{2/3}$.

The regret bound we just now obtained is in expectation over the randomized adversary strategy, and holds for any player's strategy. We now use the same line of argument as in the last part of Thm. 1's proof, to show that for any (possibly randomized) player's strategy, there exists some *deterministic* adversary strategy, with a similar expected regret bound, and with losses of bounded drift. Specifically, a result completely analogous to Lemma 5 implies that

$$\mathbb{P}\left(\mathbb{E}_{\text{player}}[R_T] \geq \frac{1}{40}\,(T-1)^{2/3}\right) \geq \frac{1}{40}\left(\frac{T-1}{T}\right)^{2/3},$$

which is at least $1/80$ for any $T > 1$ (if $T = 1$ the bound in the theorem is trivial from the non-negativity of $R_T$ for the adversary strategy that we consider). Moreover, using Eq. (4) as in the proof of Thm. 1, the probability of the loss drift being at most $\sqrt{3\log(320t)} \leq \sqrt{3\log(t) + 18}$ is at least $1 - 1/160$. Thus, the intersection of the two events is not empty, and this implies that there exists some *deterministic* adversary strategy causing expected regret $\geq \frac{1}{40}(T-1)^{2/3}$, *and* loss drift at most $\sqrt{3\log(t) + 18}$ for all $t$.

## D  Proofs of Upper Bounds

*Proof of Thm. 3.* Each loss functions equals $f_t(x_{1:t}) = \ell(x_t) + \mathbb{I}_{\{x_t \neq x_{t-1}\}}$, where $\ell_t$ is an oblivious loss function. Since the range of $f_t$ is contained in an interval of size $C$, the range of $\ell_t$ must be contained in an interval of size $C - 1$. In other words,

$$\forall x \in \mathcal{A} \;\; \ell_t(x) - \min_y \ell_t(y) \leq C - 1 \;.$$

Therefore, by definition, the range of $\ell'_t$ is contained in the interval $[0, 1]$, and the analysis of the FLL algorithm holds. Namely, if $X_1, X_2, \ldots$ is the sequence of actions chosen by FLL, then, for any $T$

$$\mathbb{E}\left[\sum_{t=1}^T \ell'_t(X_t)\right] - \min_{x \in \mathcal{A}} \sum_{t=1}^T \ell'_t(x) \;=\; \mathcal{O}(\sqrt{T}) \;, \tag{19}$$

and

$$\mathbb{E}\left[\sum_{t=1}^T \mathbb{I}_{\{X_t \neq X_{t-1}\}}\right] \;=\; \mathcal{O}(\sqrt{T}) \;. \tag{20}$$

Plugging the definition of $\ell'_t$ into Eq. (19) and rearranging terms, we get

$$\mathbb{E}\left[\sum_{t=1}^T \ell_t(X_t)\right] - \min_{x \in \mathcal{A}} \ell_t(x) \;=\; (C-1)\mathcal{O}(\sqrt{T}) \;.$$

Summing the above with Eq. (20) gives

$$\mathbb{E}\left[\sum_{t=1}^T f_t(X_{1:t})\right] - \min_{x \in \mathcal{A}} f_t(x \ldots x) \;=\; \mathcal{O}(C\sqrt{T}) \;.$$

$\square$

*Proof of Thm. 4.* Recall that $f'_t(x_{1:t}) = \frac{1}{2(C+D)}\left(\ell_t(x_t) - \ell_{t-1}(x_{t-1})\right) + \frac{1}{2}$, and note that our assumptions imply that

$$
\begin{aligned}
|\ell_t(x_t) - \ell_{t-1}(x_{t-1})| &= |\ell_t(x_t) - \ell_{t-1}(x_t) + \ell_{t-1}(x_t) - \ell_{t-1}(x_{t-1})| \\
&\leq |\ell_t(x_t) - \ell_{t-1}(x_t)| + |\ell_{t-1}(x_t) - \ell_{t-1}(x_{t-1})| \\
&\leq D + C .
\end{aligned}
$$

Therefore, $f'_t(x_{1:t})$ is always bounded in $[0, 1]$. Moreover, the action $x$ which minimizes $\sum_{t=1}^{T} f'_t(X_{1:t-1}, x)$ does not depend on the player's actions (only on $\sum_{t=1}^{T} \ell_t(x)$). Letting $x^*$ denote such an action, the standard analysis for Exp3 implies that

$$
\mathbb{E}\left[\sum_{t=1}^{T} f'_t(X_{1:t}) - \min_{x \in \mathcal{A}} \sum_{t=1}^{T} f'_t(X_{1:t-1}, x)\right] = \mathbb{E}\left[\sum_{t=1}^{T} f'_t(X_{1:t}) - \sum_{t=1}^{T} f'_t(X_{1:t-1}, x^*)\right] = \mathcal{O}(\sqrt{T}),
$$

where $X_{1:T}$ is the sequence of actions chosen by Exp3. Using the definition if $f'_t$, the left hand side above can be rewritten as

$$
\begin{aligned}
&\frac{1}{2(C+D)} \mathbb{E}\left[\sum_{t=1}^{T}\left(\ell_t(X_t) - \ell_{t-1}(X_{t-1})\right) - \min_{x \in \mathcal{A}} \sum_{t=1}^{T}\left(\ell_t(x) - \ell_{t-1}(X_{t-1})\right)\right] \\
&= \frac{1}{2(C+D)} \mathbb{E}\left[\sum_{t=1}^{T} \ell_t(X_t)\right] - \min_{x \in \mathcal{A}} \sum_{t=1}^{T} \ell_t(x) .
\end{aligned}
$$

Therefore,

$$
\mathbb{E}[R_T] = \mathbb{E}\left[\sum_{t=1}^{T} \ell_t(X_t)\right] - \min_{x \in \mathcal{A}} \sum_{t=1}^{T} \ell_t(x) = 2(C+D)\mathcal{O}(\sqrt{T}) .
$$

Using the assumption that $D_t = \mathcal{O}\left(\sqrt{\log(T)}\right)$, we conclude that $\mathbb{E}[R_T] = \widetilde{\mathcal{O}}(C\sqrt{T})$. $\qquad\square$

*Proof of Thm. 5.* First, note that, due to the bounded range and drift assumptions, $\widehat{f}_t \in [0, 1]$. Also note that

$$
f_t(x_{t-m:t}) - f_t(x \ldots x) = 2\big(C + (m+1)D\big)\big(\widehat{f}_t(x_{t-m:t}) - \widehat{f}_t(x \ldots x)\big) .
$$

As previously mentioned, we divide the $T$ rounds into $J$ consecutive epochs of the same length $T/J$, where $T/J \geq 2K(m+1)$, plus an additional final epoch of length at most $T/J$. We let $t_j$ denote the index of the first round in the $j$-th epoch. We run a mini-batched version of the Hedge algorithm [11] over the epochs: at the beginning of each epoch $j$, Hedge draws an action $X_j \in \mathcal{A}$ which is played consistently throughout the epoch. Now assume that at the end of each epoch $j$, loss estimates $g_j(x) \in [0, 1]$ for each action $x$ are available such that

$$
\mathbb{E}\big[g_j(x)\big] = \frac{1}{T/J - 2m - 1} \sum_{t=t_j+2m+1}^{t_{j+1}-1} \widehat{f}_t(x \ldots x)
$$

where the randomness used to compute each $g_j$ is independent of that used by Hedge to draw $X_j$. At the end of epoch $j$, we feed loss estimates $g_j(x)$ for each $x \in \mathcal{A}$ to Hedge. The resulting regret

can be bounded as follows,

$$\sum_{t=1}^{T} \mathbb{E}\Big[f_t(X_{t-m:t}) - f_t(x\ldots x)\Big]$$

$$\leq \sum_{j=1}^{J} \sum_{t=t_j}^{t_{j+1}-1} \mathbb{E}\Big[f_t(X_{t-m:t}) - f_t(x\ldots x)\Big] + \frac{CT}{J}$$

$$= 2\big(C + (m+1)D\big) \sum_{j=1}^{J} \sum_{t=t_j}^{t_j+2m} \mathbb{E}\Big[\widehat{f}_t(X_{t-m:t}) - \widehat{f}_t(x\ldots x)\Big]$$

$$+ 2\big(C + (m+1)D\big) \sum_{j=1}^{J} \sum_{t=t_j+2m+1}^{t_{j+1}-1} \mathbb{E}\Big[\widehat{f}_t(X_j\ldots X_j) - \widehat{f}_t(x\ldots x)\Big] + \frac{CT}{J}$$

$$\leq 2\big(C + (m+1)D\big)(2m+1)J$$

$$+ 2\big(C + (m+1)D\big)\frac{T}{J}\mathbb{E}\left[\sum_{j=1}^{J} \mathbb{E}\Big[g_j(X_j) - g_j(x)\Big|X_j\Big]\right] + \frac{CT}{J}$$

$$= 2\big(C + (m+1)D\big)(2m+1)J$$

$$+ 2\big(C + (m+1)D\big)\frac{T}{J}\mathbb{E}\left[\sum_{j=1}^{J} \Big(g_j(X_j) - g_j(x)\Big)\right] + \frac{CT}{J}$$

$$\leq 2\big(C + (m+1)D\big)(2m+1)J + 4\big(C + (m+1)D\big)\frac{T}{J}\sqrt{J\ln K} + \frac{CT}{J}\ .$$

In the last step we applied the known upper bound on the regret of Hedge with respect to losses $g_j \in [0,1]$, where $K$ is the number of actions. This is valid if, in particular, losses $g_j$ are oblivious. We now explain how to obtain oblivious estimates $g_j$ with the desired properties. At the beginning of each epoch $j$, we use the independent randomization to draw $K$ exploration steps $\{t_x\ :\ x \in \mathcal{A}\}$ from the set $T_j = \{t_j, \ldots, t_{j+1} - 2m - 2\}$ with the property that these steps are well separated. Namely, between any two $t_x$ and $t_{x'}$ there are at least $2m+1$ consecutive free time steps in $T_j$. During epoch $j$, when we arrive at step $t_x$ we freeze Hedge and play action $x_0$ for $m+1$ time steps, then we play action $x$ for $m+1$ more time steps. We use the two observed losses $f_{t_x+m}(x_0\ldots x_0)$ and $f_{t_x+2m+1}(x\ldots x)$ to compute $\widehat{f}_{t_x+2m+1}(x\ldots x)$. Because the $t_x$ are well separated, the exploration steps do not interfere with each other. Suppose now that we can draw these points such that the marginal of each $t_x$ is uniform in $T_j$. Then

$$\mathbb{E}\big[\widehat{f}_{t_x+2m+1}(x\ldots x)\big] = \frac{1}{T/J - 2m - 1} \sum_{t=t_j}^{t_{j+1}-2m-2} \widehat{f}_{t+2m+1}(x\ldots x)$$

$$= \frac{1}{T/J - 2m - 1} \sum_{t=t_j+2m+1}^{t_{j+1}-1} \widehat{f}_t(x\ldots x)\ .$$

This shows that $\widehat{f}_{t_x+2m+1}(x\ldots x)$ is a valid estimate $g_j(x)$. Moreover, for each $x \in \mathcal{A}$ the quantity $\widehat{f}_{t_x+2m+1}(x\ldots x)$ does not depend on Hedge's action $X_j$ for the current epoch $j$. It does not even depend on Hedge's past actions. Hence, Hedge is indeed run on a set of oblivious losses and the standard regret bound applies.

The last thing to prove is that we can draw $\{t_x\ :\ x \in \mathcal{A}\} \subset T_j$ such that the marginal of each $t_x$ is uniform in $T_j$. Note that giving equal probability to all well separated configurations of $\{t_x\ :\ x \in \mathcal{A}\}$ does not work, because the times steps closer to the beginning and to the end of $T_j$ appear in more configurations (for example, check the case $|T_j| = 8$ and $m = 1$). This problem can be fixed simply by arranging the points of $T_j$ on a circle, so that the first point $t_j$ follows the last point $t_{j+1} - 2m - 2$, and then enforcing well-separatedness on the circle. This makes the sample space completely symmetric, excluding those configurations of exploration points that exploited border effects.

The additional regret due to the computation of the $K$ exploration points is $2(m+1)CK$ per epoch. The final regret, including these additional costs, is then bounded by

$$2\big(C + (m+1)D\big)(2m+1)J + 4\big(C + (m+1)D\big)\frac{T}{J}\sqrt{J \ln K} + \frac{CT}{J} + 2(m+1)CKJ \,.$$

Choosing $J$ of order $T^{2/3}$ concludes the proof. $\qquad\qquad\qquad\qquad\qquad\qquad\qquad\qquad\qquad\quad$ □