[Reviews · NeurIPS 2013]

Submitted by Assigned_Reviewer_3

The paper studies the power of adaptive adversaries in full information online learning and bandit problems. More specifically, adversaries with switching cost, with memory of size 1 and with bounded memory are considered. Several matching lower bounds on the policy regret, which is a more suitable notion of regret for adaptive adversaries, are presented.

The main result is a new lower bound for bandit problems with switching costs that matches the existing O(T^{2/3}) upper bounds. This implies similar lower bounds for bandit problems with bounded memory adversaries. From this lower bound, they obtain a O(T^{2/3}) lower bound for full information problems with bounded memory adversaries that matches the existing upper bounds. The lower bound proof and the reduction from the full information to the bandit problem are interesting. Finally, when losses are stochastic and iid, authors prove a O(\sqrt{T log log log T}) regret bound for the bandit problems with switching costs with only O(log log T) switches, which is somewhat surprising.

Unfortunately, these lower bounds are proven for a larger class of problems with "bounded range" and "bounded drift". Thus, the lower bounds do not apply to the standard (and smaller) class of bounded loss functions.

The paper is well-written and most proofs seem to be correct.
Summary: The paper studies the power of adaptive adversaries in full information online learning and bandit problems. It is well-written and a nice contribution to nips.

Submitted by Assigned_Reviewer_5

The main contributions of the paper include providing lower bound of T^{2/3} for the problem of online learning with switching costs and bandit feedback. This matches the upper bound known from his problem. It is a somewhat surprising result as it is known that the full information version of this problem enjoys a sqrt{T} regret bound. Further through a novel reduction of the problem of online learning with bounded memory adversaries (under policy regret) with full information to the problems of online learning with switching costs with bandit feedback, the authors show a T^{3/2} lower bound for learning against bounded memory adversaries even in the full information case. Also it is shown that while faced with an iid adversary however one can still achieve a O~(sqrt{T}) rgeret bound with only O(log log T) switches for the switching costs bandit problem.

My main concern about the result is the O(log t) drift allowed in the lower bound. This is a bit unnatural. Further looking at where this comes from in the proof it seems like the statement might be true even with constant drift. Is it perhaps possible to get a worse lower bound of T^{2/3}/log T while keeping drift constant ? Another direction could be to try to take Xi's to be Rademacher random variables but I wasnt able to track what happens to lemma 1.

Another way to perhaps alleviate this issue might be to allow adversary to be random with the restriction that expected drift is bounded by constant which is true in your case.

None the less I find the results compelling even with the quirk of log t order drift allowed.
Summary: The paper is well-written. The results are very interesting ( and even surprising) and of definite interest to the theoretical machine learning community. I recommend the paper for publication.


Submitted by Assigned_Reviewer_6

This paper studies the problem of prediction with expert advice and finds matching upper and lower bounds on regret in terms of T (of T^{2/3}) in a number of settings: bandit feedback with adversaries of the form (bounded memory, size-1 memory, and imposing switching costs) and full-information feedback with adversaries with bounded memory.

This is a fundamental question, and it's nice to see progress here.

There are a couple of unusual things to note about the setting:
- The definition of expected regret is not the usual one. Traditionally, one looks at the expected difference between the costs of the selected actions and the performance of the best fixed action in hindsight (meaning given the actual history of play). Instead, they look at the expected value of the difference between the actual performance and that of the best fixed action (not taking the actual history into account, but rather assuming that the fixed action had been played all along). This difference is salient because they consider adaptive adversaries who e.g. take into account a limited history when selecting their cost functions. The regret definition used here ("policy regret") seems to be the right one for these settings, and the paper mostly does a good job of making this distinction. But since this is a potential source of confusion, I would have preferred if the paper had been more explicit with references to the cases where similar results are known under the standard expected regret. Also, it's strange that Table 1 summarizing results doesn't provide standard references for the results in prior work.
- The results require a weakening of the usual assumption of bounded loss values (and hence the reproving of the corresponding upper bounds, which the paper does do). I still think the results are interesting, but of course it would be nicer to have results in the standard model.

The paper gives a thorough treatment, with nice proof sketches of the main results in the main body of the paper, and a nice discussion of open questions.
Summary: This paper makes progress on interesting fundamental questions, and is presented well.
Author Feedback

Author rebuttal: We thank the reviewers for their comments and suggestions.

Assigned_Reviewer_5: Thanks for the suggestions regarding constant drift. We agree that it would be more aesthetic if the drift were constant, but we currently don’t know how to make it work. The difficulty is to show that the player doesn’t get any information by staying on the same action, and only gets a little bit of information by switching actions. We know that replacing the Gaussian random walk with a Rademacher random walk doesn’t work, but we will definitely keep thinking about the other suggestions.

Assigned_Reviewer_6: We will try to make the distinction between standard regret and policy regret as explicit as possible throughout the paper. We will add references to prior work in Table 1 where appropriate.